# A Long-Term Trend Study of Tuberculosis Incidence in China, India and United States 1992–2017: A Joinpoint and Age–Period–Cohort Analysis

**DOI:** 10.3390/ijerph17093334

**Published:** 2020-05-11

**Authors:** Yiran Cui, Hui Shen, Fang Wang, Haoyu Wen, Zixin Zeng, Yafeng Wang, Chuanhua Yu

**Affiliations:** 1Department of Epidemiology and Biostatistics, School of Health Sciences, Wuhan University, Wuhan 430071, China; 2019283050055@whu.edu.cn (Y.C.); shenhui1127@whu.edu.cn (H.S.); wangfang0923@whu.edu.cn (F.W.); haoyuwen@whu.edu.cn (H.W.); 2019283050036@whu.edu.cn (Z.Z.); wonyhfon@whu.edu.cn (Y.W.); 2Global Health Institute, Wuhan University, Wuhan 430071, China

**Keywords:** tuberculosis incidence, joinpoint regression analysis, age-period-cohort effect, trends

## Abstract

Tuberculosis (TB) is one of the major infectious diseases with the largest number of morbidity and mortality. Based on the comparison of high and low burden countries of tuberculosis in China, India and the United States, the influence of age-period-cohort on the incidence of tuberculosis in three countries from 1992 to 2017 was studied based on the Global burden of Disease Study 2017. We studied the trends using Joinpoint regression in the age-standardized incidence rate (ASIR). The regression model showed a significant decreasing behavior in China, India and the United States between 1992 and 2017. Here, we analyzed the tuberculosis incidence trends in China, India, as well as the United States and distinguished age, period and cohort effects by using age-period-cohort (APC) model. We found that the relative risks (RRs) of tuberculosis in China and India have similar trends, but the United States was found different. The period effect showed that the incidence of the three countries as a whole declines with time. The incidence of tuberculosis had increased in most age group. The older the age, the higher the risk of TB incidence. The net age effect in China and India showed a negative trend, while the cohort effect decreased from the earlier birth cohort to the recent birth cohort. Aging may lead to a continuous increase in the incidence of tuberculosis. It is related to the aging of the population and the relative decline of the immune function in the elderly. This should be timely population intervention or vaccine measures, especially for the elderly. The net cohort effect in the United States showed an unfavorable trend, mainly due to rising smoking rates and the emergence of an economic crisis. Reducing tobacco consumption can effectively reduce the incidence.

## 1. Introduction

Tuberculosis (TB) is a chronic infectious disease caused by Mycobacterium tuberculosis infection and the main cause of a single infectious disease. It is listed as one of the ten most important causes of death from infectious diseases in the world [1]. It has become a death-related infectious disease at a higher level than human immunodeficiency virus (HIV). China has the world’s second-largest TB epidemic, behind India, with more than 1.3 million new cases of TB every year. India has an estimated 2 million new cases every year, accounting for 18% of the estimated burden of TB in the world [2]. The number of TB cases in the United States decreased from 1992 to 2017 [3]. However, the TB incidence in 2009 showed a lowest recorded rate in the United States. There was a large decrease of reported TB cases during 2009, which might have resulted from improving TB control [4,5].

Most of the studies on tuberculosis in China and the United States are based on the age-period-cohort framework of mortality [6,7], whereas some studies focus on the risk factors of tuberculosis [8]. But few studies emphases on the incidence of tuberculosis. In this regard, a sophisticated study was conducted on the current manuscript to outline incidence status and trend of TB in China, India, and the United States from 1992 to 2017.

## 2. Materials and Methods

### 2.1. Data Sources

The data of this study mainly come from the estimated burden of TB disease in the “TB management information system”, which annually reports on global TB control (1990–2017), GBD 2017. TB data for three representative countries like China, India, and the United States can be obtained from the Global Health Data Exchange (GHDX) website of the Institute for health metrics and evaluation IHME (http://ghdx.healthdata.org/gbd-results-tool). GBD 2017 provides a comprehensive upshot for the 282 causes of death in 195 countries and territories from 1990 to 2017 [9].

### 2.2. Statistical Analysis

#### 2.2.1. Age–Period–Cohort Analysis

APC analysis has a unique ability to succinctly describe the complex situation of the social, historical and environmental factors that affect individual groups at the same time. Therefore, it is widely used to solve problems of importance to social change, disease causes, aging, population process and dynamic research. Although there are a lot of studies on infectious disease trends in China, we can also separate the effects of different ages, periods and cohorts to analyze TB incidence on the research outcomes [10,11,12]. It is difficult for traditional statistical methods to eliminate the interaction of age, period and cohort effect. The Intrinsic Estimator (IE) as a new method of coefficient estimation [13]. The model expression of APC is generally written as
$$Y=\log\left(M\right)=\mu +{\alpha \mathrm{age}}_{1}+{\beta \mathrm{period}}_{1}+{\gamma \mathrm{cohort}}_{1}+\varepsilon$$
where, M stands for the incidence of the corresponding age group, μ stands for the intercept item, α, β, and γ stand for the corresponding age, period and cohort effect, and ε is the random error. It has age_1_ = period_1_ − cohort_1_ [14].

Data for TB rates from 1990 to 2017 in ages 20 to 79 for specific three countries were extracted from the GBD. To calculate age–specific TB incidence by birth cohort, we classified patients into 18 cohorts of birth (i.e., 1903–1912; 1913–1917; 1918–1922; 1923–1927; 1928–1932 and so forth until 1988–1992) on the basis of available data by age. We estimated age–adjusted TB incidence according to 5 calendar periods of 5-year intervals each (1992–1997, 1997–2002, 2002–2007, 2007–2012, and 2012–2017), and 12 age groups (20–24, 25–29, 30–34, 35–39, 40–44, 45–49, 50–54, 55–59, 60–64, 65–69, 70–74, and 75–79 years).

Particularly, the analysis related to APC is accessible via StataCorp, College Station, TX, USA. Furthermore, we have categorically carried out the Wald test peculiarly based on the outcomes of the APC model, due to the fact that the corresponding value of *p* < 0.05, which furnishes a vivid significance statistically. Moreover, the deviation test in addition to the Akaike information criterion (AIC), and Bayesian information criterion (BIC) was utilized to quantify and analyze the extent of fitting of the model.

#### 2.2.2. Joinpoint Regression Analysis

The incidence rate of TB can be estimated by the Joinpoint Regression Software. The software was used to estimate the time trend of TB incidence rate and determined the changes in different years in China, India and the United States based on the breakpoints. That is to say, the model can make us see the trend of the tuberculosis incidence rate and find the change point of each trend intuitively, which is very suitable for piecewise regression. In the regression analysis, the annual percentage change (APC), the average annual percentage change (AAPC) and the 95% confidence interval (CI) were obtained [15]. This analysis was performed through the Joinpoint regression program version 4.6.0.0 (April 2018) from the Surveillance Research Program of the U.S. National Cancer Institute.

## 3. Results

### 3.1. The Overall Trends in TB Incidence in China, India, and the United States

Figure 1 and Figure 2 provide the characteristics curves that representing the temporal evolution of the crude incidence rates (CIR) and the age-standardized incidence rates (ASIR) of TB in China, India and the United States during the period from 1992 to 2017. CIR of TB in China is observed to have a significant decrement from 1992 to 2017 as compared to the United States. During1992–1993, it remained unchanged, but it dropped significantly from 1993 to 1994 in the United States. Furthermore, the TB incidence of in India is higher than in China and the United States from 1992 to 2017. Similarly, as observed in Figure 2. China’s ASIR steeply decreased from 147.65 in 1992 to 54.18 in 2017 per 100,000 persons. Likewise, India’s ASIR dropped by 28.06%, decreased from 299.56 in 1992 to 215.51 in 2017 per 100,000 persons. In contrast, the United States’ ASIR fell slowly from 5.33 to 2.34 per 100,000 during the same period.

### 3.2. Descriptive Analysis of Gender of TB Incidence in China, India, and the United States

Age-standardized incidence rate (ASIR) for men and women at all ages for TB from 1992 to 2017 is described in Figure 3. The standardized incidence of tuberculosis in men and women in three countries has declined from 1992 to 2017 but among men is higher than women. The incidence of TB in the United States experienced a slight increase by 5.04% from 1992 to 1998 in men and a decrease by 52.48% from 1992 to 2017 in women.

### 3.3. Trends in Age-Standardized TB Incidence Rates Using Joinpoint Regression Analysis

The Analysis by Table 1 showed that APC and AAPC of TB incidence in China, India and the United States from 1992 to 2017. The TB incidence rate in China between 1992 and 2017 ranged from 147.65/100,000 to 54.18/100,000 population. The regression model showed a significant decreasing behavior in China between 1992 and 2017 (APC: −4.52; 95%CI: −4.8 to −4.2; *p* < 0.001). At the same time the regression model also showed a downward trend in India between 1992 and 2017 (APC: −1.23; 95% CI: −1.3 to −1.1; *p* < 0.001), which decreased from 299.56/100,000 population in 1992 to 215.51/100,000 population in 2017. The regression model also indicated a decreasing behavior between 1992 and 2017 in the United States (APC: APC: −3.71; 95% CI: −4.0 to −3.4; *p* < 0.001), APC had increased by 0.36% during 1992–1995. Trend analysis of TB incidence rate decreased by 4.31% during 1998–2017.

### 3.4. The Variation in Age, Period and Cohort on TB Incidence

In order to show an age-related trends of TB from 1992 to 2017, with an interval of 5-years in the context of different age groups, is provided in Figure 4. The age-specific TB incidence rate showed a similar behavior for the same age group in both China and India. Particularly, in 2017, the TB incidence rate in China is relatively low, rising from 63.27 per 100,000 in the age group 20–24 to 106.95 per 100,000 in the age group 75–79. TB incidence rate in India has presented a sharp increase from 215.65 per 100,000 in the age group 20–24 to 439.42 per 100,000 in the age group 75–79, and we can see that this significant trend is more frequent in the older age groups. The highest TB incidence rate was observed in India as comparing to other countries. In contrast, the TB rate in the United States has increased marginally for the age group 20–69, but it has continued to decrease in other age groups. In 1997, the United States reached the highest level in the 65–69 age group (11.95 per 100,000 people). Overall, in the United States, an inclusive age-specific TB rate has remained steady, but there is a crest in the 65–69 age group in each period.

As shown in Figure 5, the birth cohort of China, India and the United States fluctuated significantly. It can be easily felt that for the same age group, the incidence of tuberculosis has a sharp downward trend in China and India. While in the United States, the incidence of tuberculosis in the 65–69 age group usually rises. We found that the influence of age and cohort on the incidence of tuberculosis cannot be presented independently. Therefore, we determined that the APC model has the most appropriate design based on available data.

### 3.5. The Age, Period, and Cohort Effects on TB Incidence

#### 3.5.1. Age Effect

Figure 6 shows the risk ratios (RRs) of TB incidence in China, India and the United States. The risk of TB incidence was found to have decrement in the 65–69 age group. The situation in China and India was similar from lower to upper age groups. The risk of TB incidence continued to increase for the 35–60 age group, where the highest level was in the 75–79 age group. Later on, the RRs smoothed at that level for the 60–69 age group. Therefore, the older age groups have a higher risk of TB incidence than younger ones. On the contrary, the risk of TB incidence in the United States had an obvious increase for the 20–35 and 60–65 age groups in Figure 5, but it showed a downtrend for the 45–60 and 65–79 age groups. Among them, the age RRs of 70–74 years decreased by about 20%. It’s clear that the age effect on TB incidence of the United States was very different from that of China and India.

#### 3.5.2. Period Effect

The periodic RRs of TB incidence in China, India and the United States are plotted in Figure 7, which demonstrated that RRs had similar declining trends for incidence in three countries. However, In China, the RRs of incidence fell markedly to 0.43 in 2017. Similarly, In India, the RRs of incidence dropped down to 0.85 in 2017. Interestingly, the periodical effect of TB incidence first reached the highest point, and then fell to a lower value in the United States. So the RRs of the United States had a declination to about 0.41 in 2017, but reached the uttermost value of about 1.13 in 1997.

#### 3.5.3. Cohort Effect

Similar to the above, the cohort RRs of TB incidence in China, the United States and India are displayed in Figure 8. But these patterns were found to have the opposite behavior. In terms of TB incidence in China, the risk increased before 1942, and then declined after 1943, and it had risen slightly since 1982. The overall curve showed a downward trend with a different rate of decrease in India. The cohort RRs of TB incidence in the United States between 1908 and 1957 years showed a declining trend, while it increased nearly twice from 1957 to 1997, with it being the largest increase. The trend embodied that in the United States, the risk of tuberculosis in humans before 1957 was lower than that of future generations. The APC model analysis results of TB incidence in China, India and the United States from 1992 to 2017 are shown in the Table A1.

## 4. Discussion

The results of the APC model show that the incidence in China and India decreases with age and cohort, but increases with age, while the incidence in the United States rises first and then decreases with age and period, but declines and then rises with the cohort. Similar dominion studies from Hong Kong described the application of APC modeling in the study of tuberculosis incidence [16]. Although APC models are often used to study trends in chronic diseases, they can also be used to analyze infectious diseases as well [17,18]. As the trend of tuberculosis is meticulously related to age and period [19]. Therefore, we analyzed the long-term trends in TB incidence using the APC model and to explore the age, period, and cohort effects, the following is a further analysis of the background, causes and influencing factors of different trends.

### 4.1. Age Effect

In addition to the other cumulative factors, human age has a prominent relationship with the adopted disease. Age RRs in all three countries has a high impact on tuberculosis. Moreover, the risk factors are closely related to the age effect. The prominent effect of age RRs is found over the age of 25 years, which is mainly attributable to malnutrition [20]. With the increase of age, the elderly will be affected by risk factors such as diabetes [21] and indoor air pollution. The high infectivity of tuberculosis affects the vulnerable population more. The immune function of the elderly over 60 years old will decline [22], which will lead to diabetes and other diseases. It follows that diabetes is a high-risk factor for tuberculosis [23].

Specifically, we found it to increase with the increase of age, because the population aging reached the highest in the age range of 65–69. Most elderly people with tuberculosis in China lived longer because their life expectancy increased by 7.79 years from 1990 to 2015 [24]. Onwards from 1992 to 2017, the proportion of the elderly aged over 65 in China increased from 6.2% to 11.4%. In the next 30 years, the global population aged 65 and over will grow to 16%. According to an estimation, by 2050, China’s population corresponding to age 65 or over will be 400 million [25]. With the rapid aging of China’s population, the prevention and control of tuberculosis in the coming future will face greater challenges. Due to immune decline, physiological changes, malnutrition, and many other reasons, the older generation is easily suffering from a variety of diseases, which can also increase the risk of tuberculosis. The elderly are seriously affected by infectious diseases, so we should strengthen the screening and treatment of the particular age group in specific [26]. TB burden in the senior citizens (60+ years of age) is particularly perceived to be higher than in other groups during the time duration from 1992 to 2015 [27].

In India, the age effect of the 35–60 age group continued to increase, with a high increase rate of 0.466 and was instigated to decline slightly in the 60–70 age group. For example, alcohol after using, alcohol dosage and alcohol-related problems as risk factors for tuberculosis incidence [28]. First of all, drinking alcohol can damage the immune system, thus increasing the susceptibility of tuberculosis and the reactivation of potential tuberculosis [29]. As expected, the incidence of alcohol-induced TB increases with the incidence of TB [30]. All three factors are associated with an increased risk of tuberculosis. In addition to alcohol consumption as a risk factor for tuberculosis in India [31], the age RRs of increase is mainly due to the escalation of alcohol ingestion in India. Alcohol consumption is the main cause of tuberculosis, with the most severe impact in Africa [30]. That is to say, the rapid rise of age RRs in India in the 35–60 age group is related to alcohol consumption.

However, in the United States, we can see that there are two peaks in TB corresponding to 35–39 and 65–69, respectively. Our research results are consistent with the Shareen A. Iqbal study [32]. The trend of the age effect is different, the age RRs of tuberculosis in the old public is high in the United States [33]. Because there are many basic diseases such as diabetes in the elderly, over 60 years old, and low immunity will easily leads to tuberculosis. And moreover, the 20–35 and 45–64 age groups are associated with simple extrapulmonary tuberculosis [34]. But it has present a significant decline after the age group of 65–69 years old. People, >65 years are associated with most types of extrapulmonary tuberculosis [35], which indicated that age effect has a great influence on the incidence of tuberculosis. However, the United States as a whole is in high age RRs of TB.

### 4.2. Period Effect

From the available facts, the period RRs of incidence of TB for the time duration of 1992–2017 has been seen to have a downward trend in China. The possible reasons for the declining RRs in China may be the enhancement of living standards, development of health services, progress of science and technology, and the monitoring of infectious diseases. With the economic development of China’s reform and opening up, coupled with the continuous improvement of the national basic medical and health reform measures and medical technology, people’s living conditions have been greatly improved. In 2005, China rebuilt public health service facilities and formulated a new five-year plan for tuberculosis prevention and treatment [36]. In this regard, an indication was given by Premier Wen Jiabao, speaking at the National People’s Congress in March 2006, that public health is a key component of the country’s 11th 5-year development plan is very encouraging. He highlighted the need to improve rural and urban health services, making them affordable for all, and specifically mentioned the need to control HIV/AIDS, tuberculosis [37]. Explicitly during the time span 1992–2017, the RRs had declined all the time, the reason is not only due to the gradual reform of China’s health care system in 2009 [38], but also to the continuous adoption of effective measures to reduce the impact of air pollution on tuberculosis in the future [39].

Similarly, for the same retro, the period RRs in India declined from 1992 to 2017. In 2012, the WHO reported that the incidence of tuberculosis in all age groups in India was 170 cases per 100,000 people per year, lower than our research results [40]. In 1993, India introduced a new policy to control TB, and the TB control plan in India has successfully improved the possibility of treatment, which can avoid 200,000 deaths [41]. A total of 2.8 million people in India were suffering from tuberculosis, which accounts for a quarter of the world’s TB burden and makes it difficult for period RRs to decline in 2016 [42]. From that time on, India’s TB control program has reduced the incidence of TB and prevented its further spread.

On the other hand, the decrement in the RRs of TB may be due to improved medical measures [33], despite the fact that the United States is a developed country with rapid economic development. In contrast, from 1992 to 1997, period RRs in the United States showed an upward trend, unimproved indices in program correlated with increases in period RRs of TB incidence from 1992 to 1997 of the study period [43]. Because the high smoking [28] rate and high drinking [31] incidence rate, as the risk factors of tuberculosis, will not only damage the immune system, but also lead to health loss [44]. During the Gulf War, the influx of immigrants into the United States led to an increase in the incidence of tuberculosis. Analysis of TB surveillance data in the United States from 1993 to 1998 showed that the number of TB cases among US immigrants increased by 2.6% [45]. In recent years, the United States has increased TB surveillance, with the lowest incidence in 2017 [5]. However, the United States is still aimed to take miles in a particular direction to further prevent the spread of tuberculosis, so as to achieve the goal of eliminating tuberculosis forever.

### 4.3. Cohort Effect

The cohort effect is an important aspect. Generally, the cohort RRs differences in the birth cohort refer to poise between new infections and weakened immune responses to previous infections [46].

The cohort RRs showed that the turning point of the cohort effect around 1920 probably imitated the events in southern China. What is striking, is that the rise of the turning point was interpreted as the reflection of the turbulence related to the demise of the Qing Dynasty in the whole population range, which was caused by the increase of population flow and the exposure of the baby groups born in that period. We found that the duration grasped its highest point between 1938 and 1942, and this cohort was related to Japan’s invasion of China. The “Lugou Bridge Incident” initiated by Japan in 1937 was considered as historical mutilation of and humiliation to China, which provoked a comprehensive war of aggression against China and set off a prelude to the entire nation’s resistance to Japan. The war not only instigated the deterioration of the living environment and the destruction of medical and health services but also prevented their lives and health from being assured and safe. The same scenario was persistent till 1945, and later on, the overall risk of the birth cohort decreased. Predominantly, during the period of accelerated decline, China established a medical security system just after the People’s Republic of China came into being [47]. Besides, China established a medical insurance system for the employees, including the public medical system and the labor insurance medical system, keeping in view the instance of the Soviet Union [48]. Additionally, it exercised the gradual improvement of the medical insurance system for urban employees and inaugurated a rural cooperative medical system. After the Second Revolution, China launched a patriotic public health campaign to reduce the risk of tuberculosis [49]. With the continuous development of society and the economy, the popularization of health education knowledge, the enhancement of people’s health awareness and the improvement of nutritional status, to a certain extent, reduce the risk of births.

From 1978 to 1987, the downward trend was slow or even upward. This was due to China’s exploratory reforms. Since 1978, the re-development of healthcare has also brought some disadvantages. Market-oriented reforms have disrupted the public health insurance system and health provision system [50]. Basic health services cannot be afforded, and medical poverty is caused by high medical expenditures, and widening gaps in health status in China [51].

The Cohort effect on the incidence of tuberculosis showed a decreasing trend in India from 1908 to 1962 and from 1987 to 1997. But there was seen a slight upward trend of cohort RRs in India amongst between 1962 and 1987. Because India is a country with a high burden of TB, and TB is a disease mainly originated to poverty. Additionally, India falls ranked second in the context of the human population after the largest populated country, China, globally and a large population in India is still at a high risk of contracting TB [52]. Food supply is difficult to meet the needs of the population, which will lead to malnutrition in most Indians [53]. With the rapid development of urbanization, the population is increasing rapidly, and the prevalence of tuberculosis is also increasing in India along with the most crucial aspect of TB risk factors [54]. The most important thing in India is the risk factors for TB. Besides smoking and drinking, AIDS and diabetes are related to the rise of cohort RRs in India [55]. However, cohort RRs rapidly declined from 1992 to 1997, which signifies that India has taken effective measures to reduce the burden of tuberculosis. Predominantly, in the 1990s, the authorities, professionals, and institutions were given special considerations to strengthen TB control and track closely the associated gears, for a better future [56].

In the case of the United States, we determined that the cohort RRs of TB largely decreased steadily prior to 1957, but the same trend was not existed n younger cohort people [46,57]. Tuberculosis cases are significantly reduced but are highly contagious. In the period 1900–1940, the United States tried to use specific immunization and treatment methods to prevent the occurrence of tuberculosis [58]. From a perspective of the Cold War, the United States suffered an economic crisis in 1953, and it is per capita income fell rapidly. At the same time, the Vietnam War broke out in 1955. These factors intricate a large number of financial catastrophes, which increased the risk of death factor in the United States [6]. Due to these factual conditions, the RR of this cohort was the lowest from 1953 to 1957, but since 1958 it has grown at a relatively high rate. Although the 1958–1997 cohort RRs have been on the rise, due to structural changes in TB in the health sector, further evaluation of the effectiveness of TB is needed to accelerate the decline of TB in the United States [59,60,61]. The increase of RR in the United States since 1958 was mainly related to the upsurge of the smoking rate. Though smoking was seen to be one of the most dangerous factors in the United States in the 1960s as well [62].As we all know, Tuberculosis and smoking had a history of more than 100 years [63], and the relationship between smoking and tuberculosis was proposed a long time ago, but recently it has been confirmed [64,65,66]. Smoking rates are also increasing in the United States, and 13% of TB patients in the world are caused by smoking every year [67]. Non-U.S. born individuals account for an increasing proportion of total TB cases in the United States. In the United States, the incidence of tuberculosis is increasingly concentrated in non-U.S. born people [68]. The reason for the rise of cohort RRs in the United States should be related to the size and composition of the immigrant populations. In the 1970s, the burden of tuberculosis was increased by European immigrants [69]. In the period 1993–1997, cohort RRs in the United States were close to 1, part of the reason is being the tuberculosis outbreak the early 1990s of some low incidence areas [70].

This article also has certain limitations. Because it is based on GBD 2017 data, there is still a certain deviation in the completeness and accuracy of the TB incidence data we use. This article does not analyze different types of tuberculosis subgroups. We will further analyze the different subgroups of tuberculosis in order to obtain more accurate analysis results.

## 5. Conclusions

In summary, in terms of the age effect of tuberculosis, China, India, and the United States generally show an upward trend, but the United States shows a significant downward trend after the age of 65. It can be said that age is an important factor affecting tuberculosis. In addition to the aging population, the incidence of tuberculosis is also related to the risk factors of tuberculosis. Among people born in the United States, the incidence of tuberculosis first declines and then rises. This is related to the national policy background and whether there is a war. Effective intervention should be carried out in time, especially in the elderly men who smoke or have diabetes.

## Figures and Tables

**Figure 1 ijerph-17-03334-f001:**
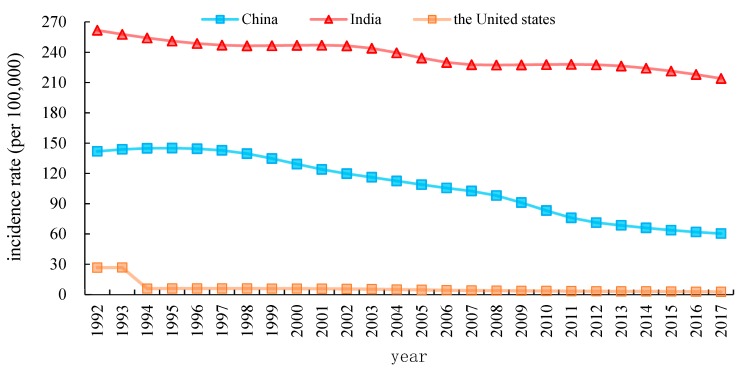
Trends in crude incidence rate (CIR) from tuberculosis (TB) in China India and the United States, 1992 to 2017.

**Figure 2 ijerph-17-03334-f002:**
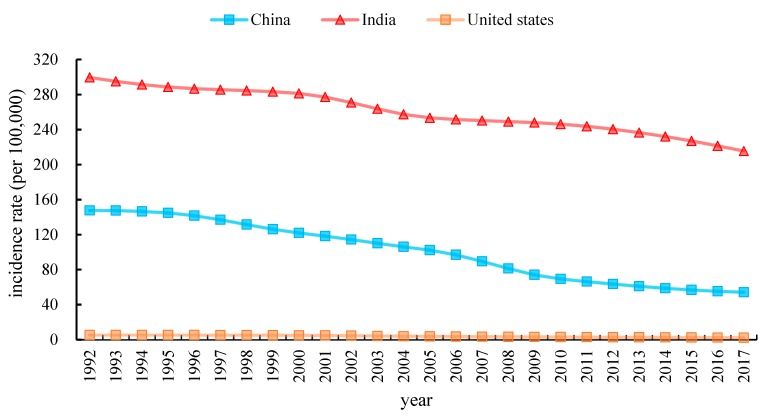
Trends in age-standardized incidence rate (ASIR) from TB in China, India and the United States. 1992 to 2017.

**Figure 3 ijerph-17-03334-f003:**
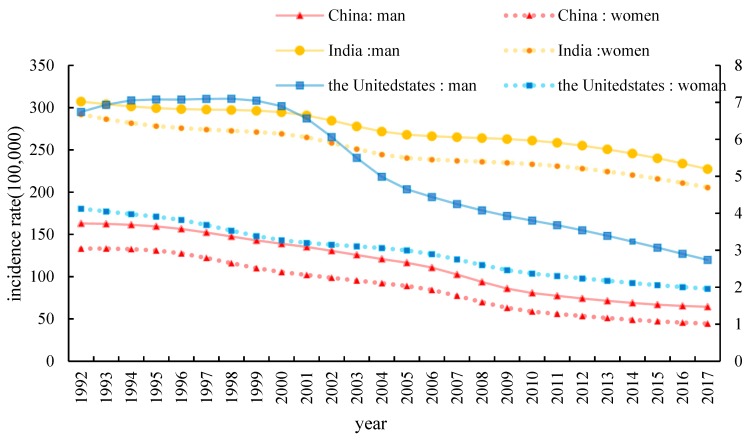
Trends in age-standardized rates for tuberculosis in men and women from 1992–2017 in China, India, and the United States.

**Figure 4 ijerph-17-03334-f004:**
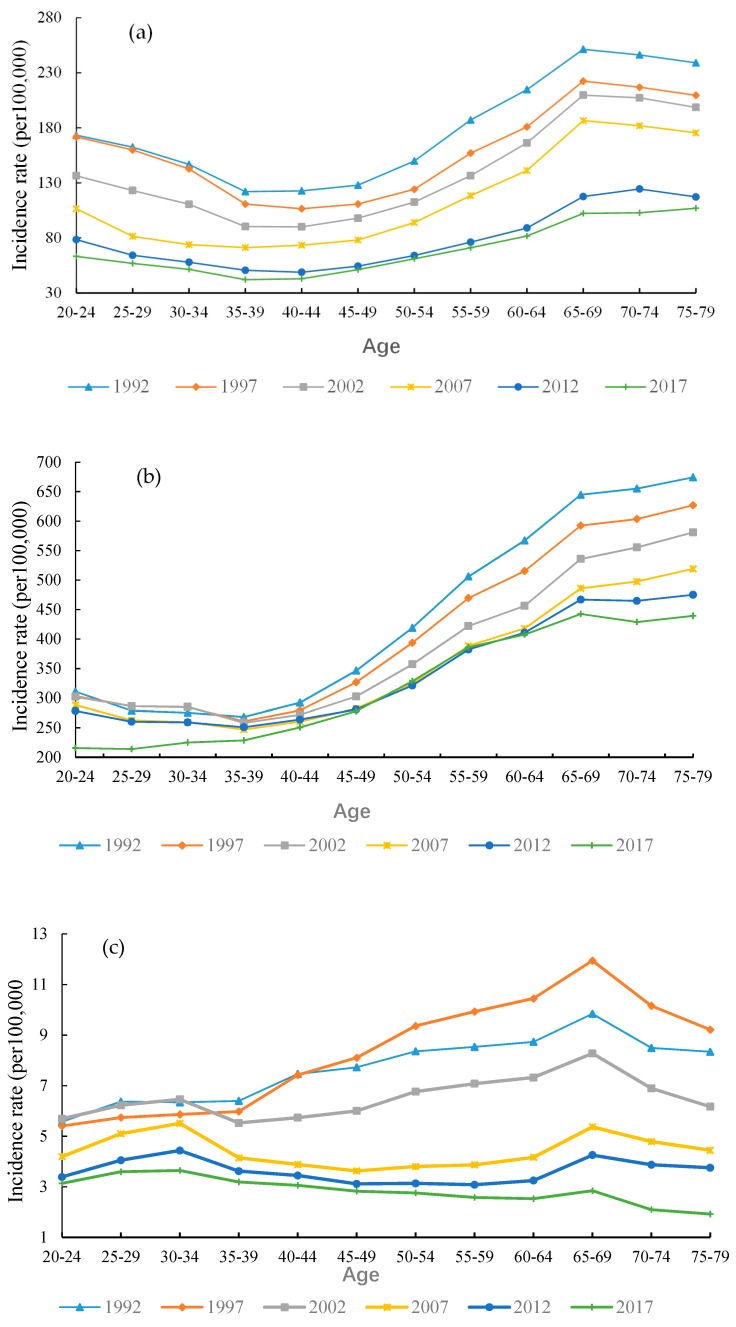
Age-specific TB incidence rate for (**a**) China, (**b**) India, (**c**) United States, 1992 to 2017.

**Figure 5 ijerph-17-03334-f005:**
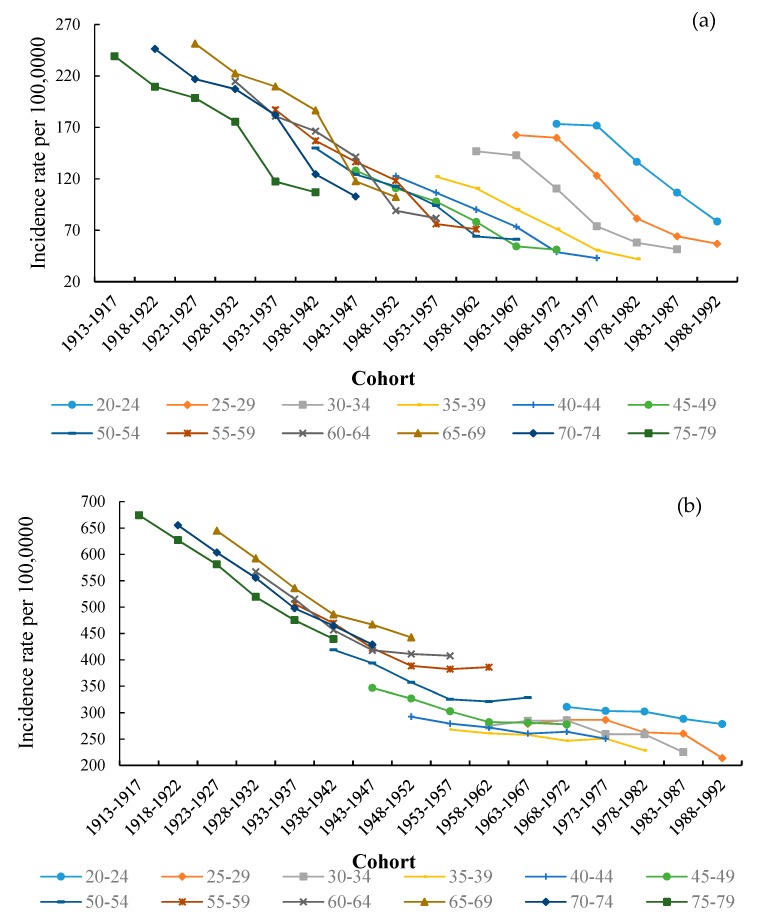
Cohort-based variation in age-specific TB incidence in (**a**) China, (**b**) India, (**c**) United States.

**Figure 6 ijerph-17-03334-f006:**
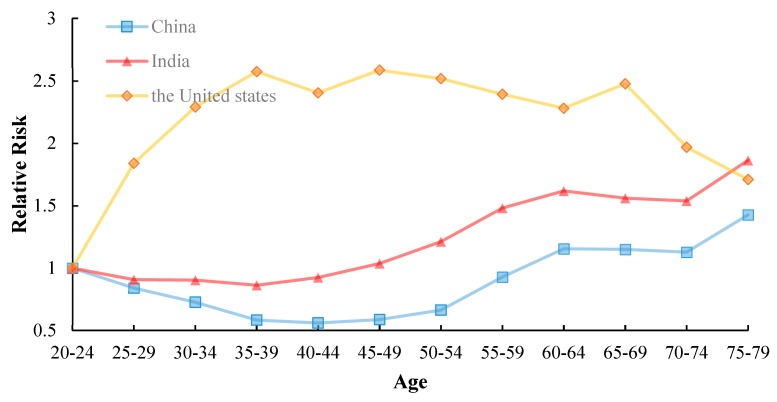
Age effects on TB incidence rate for China, India, United States, from 1992 to 2017.

**Figure 7 ijerph-17-03334-f007:**
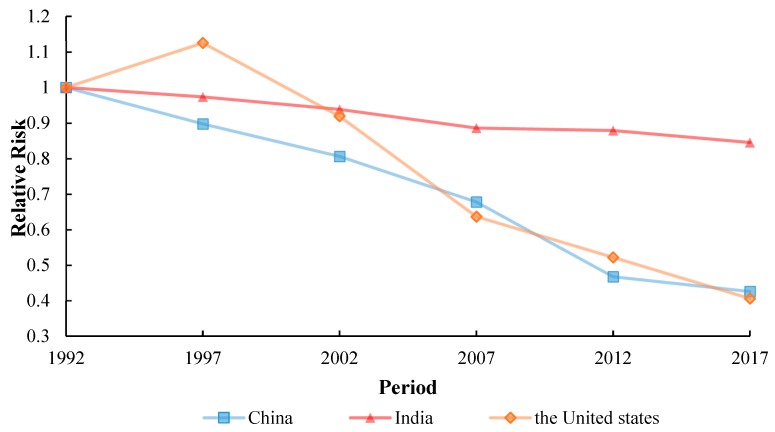
Period effects on TB incidence rate for China, India, United States, from 1992 to 2017.

**Figure 8 ijerph-17-03334-f008:**
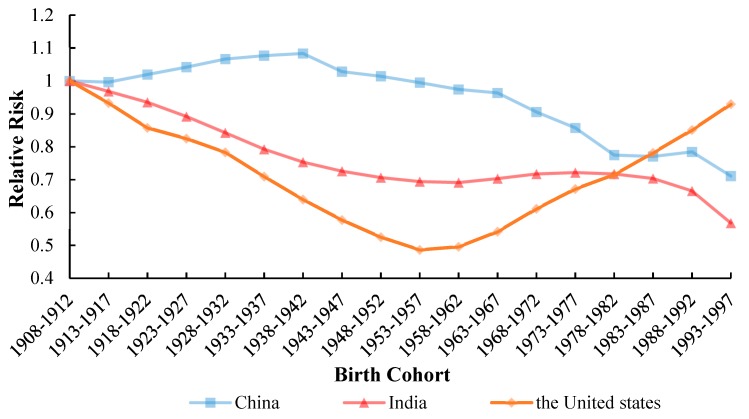
Cohort effects on TB incidence rate for China, India, United States, from 1992 to 2017.

**Table 1 ijerph-17-03334-t001:** Trends in tuberculosis incidence in China, India and the United States, 1992–2017.

Segments	China		India		The United States	
	Year	APC * (95% CI)	Year	APC * (95% CI)	Year	APC * (95% CI)
ASIR						
trend1	1992–1995	−0.3 (−0.9, 0.2)	1992−1995	−1.3 * (−1.5, −1.0)	1992–1995	0.4 (−0.5, 1.2)
trend2	1995–2006	−3.5 * (−3.6, −3.5)	1995−2000	−0.4 * (−0.6, −0.2)	1995–2000	−1.3 (−1.8, −0.8)
trend3	2006–2009	−9.2 * (−10.1, −8.3)	2000−2005	−2.1 * (−2.3, −1.9)	2000–2005	−5.6 * (−6.1, −5.1)
trend4	2009–2013	−4.8 * (−5.3, −4.3)	2005−2008	−0.6 * (−0.7, −0.5)	2005–2012	−4.4 (−5.9, −2.9)
trend5	2013–2017	−2.8 (−3.1, −2.5)	2008−2017	−2.3 * (−2.4, −2.1)	2012–2017	−3.7 (−3.8, −3.6)
AAPC *	1992–2017	−4.0 * (−4.1, −3.8)	1992−2017	−1.3 * (−1.4, −1.2)	1992–2017	−3.2 * (−3.4, −3.0)

Note: *, Indicates that the Annual Percent Change (APC) is significantly different from zero at the alpha = 0.05 level; APC, annual percentage change; AAPC, average annual percent change; CI, confidence interval; ASIR, age-standardized incidence rate.

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
