# Peer review of "A Long-Term Trend Study of Tuberculosis Incidence in China, India and United States 1992–2017: A Joinpoint and Age–Period–Cohort Analysis"

_ijerph, 2020, doi:10.3390/ijerph17093334_

Round 1

Reviewer 1 Report

An analysis like this can reveal important insights as the pioneer work in this field by Wade Hampton Frost (should be cited, and certainly read) shows. However, this paper seems to generated by some translation program and is in parts totally incomprehensible. Conclusions are formulated in flowery language but provide little information that can be gleaned from the data. Finally, as TB incidence, in many settings, has many sex differences, analysis should be carried out by sex.

Author Response

The uploaded Word is my revised manuscript, please check it. Thanks to the moment of reviewer.

Reviewer 2 Report

The work has a very complex statistical scaffolding with very strict formal sequences. The authors used the Joitpoint regression over time to identify several points and trend variations for TB in three States (time series). However, it is not clear what the application of the work is, what its purpose, its impact and practical aspects are in epidemiological terms. It would not be bad to point out, in a passage, its possible limitations, for example, that the work focuses on theoretical aspects that can have marginal practical consequences. The bibliographic part seems rather solid. Overall, the work deserves to be published with the warnings given above and taking into account the following minor suggestions for change.

Some minor concerns

###Abstract: For readers, please specify the acronym GBD, what’s GBD (line 18)

####Abstract, lines  24-25: include results for India, China and USA in terms of  RRs

#### Lines 45-47: Report TB cases for USA

#### The authors stated “…so as to provide a vivid scientific origin for an intricate control and prevention of Tuberculosis….line 53-54, specify the scope better

### line 86  “on the on the” amend

###Tuberculosis is a disease with several subgroups, it should be specified whether this clinical definition (TB) covers specific forms of tuberculosis such as pulmonary TB, or other types, or all, explain

### It is not clear, according to which criterion the age classes have been defined( …we classified patients into 18 cohorts of birth), more information is required It is an application and/or possibility allowed by Jointpoin analysis?

Author Response

Thanks to the moment of the reviewer. The uploaded Word is my revised manuscript, please check it.

Reviewer 3 Report

This study is well conceptualized and realized. I propose to insert the infection containment measures adopted over the years in the three countries to achieve a reduction in the incidence of TB (such as: vaccination campaigns or others).

In my opinion it needs to find other explanation for the observed increased incidence in U.S. from 1992 to 1997, especially in young adults  (return from Gulf War?).

Furthermore, the prevalence of AIDS patients in the three countries since the 1980s should also be analyzed and compared with TB data.

Author Response

Thanks to the comments of the reviewer. The uploaded Word is my revised manuscript, Please check it.

Round 2

Reviewer 1 Report

I very much appreciate the way my comments have been addressed.  Yet, I think your ms can be further improved. Language can be improved further and the ms would also be improved by reducing the amount of description. A lot of text simply describes what can be seen in the graphs without ant real "analysis". My suggestion is to reduce the amount of text to information not provded by the (nice) graphs. 

Author Response

Dear reviewer,thank you for your questions about my article, including the prominent marks made, thank you very much for your attention and recognition. In response to the problem of too many description classes you mentioned, I made corrections and general deletions on the description data of the main body of the article. Each section has been adjusted. I hope you can be satisfied.

Round 3

Reviewer 1 Report

no more comments